# Cytokinin and Metabolites Affect Rhizome Growth and Development in Kentucky Bluegrass (*Poa pratensis*)

**DOI:** 10.3390/biology12081120

**Published:** 2023-08-11

**Authors:** Fu Ran, Xiaoming Bai, Juanxia Li, Yajuan Yuan, Changning Li, Ping Li, Hui Chen

**Affiliations:** 1College of Grassland Science, Gansu Agricultural University, Lanzhou 730070, China; ranfu1907@163.com (F.R.);; 2Key Laboratory of Grassland Ecosystem, Gansu Agricultural University, Lanzhou 730070, China

**Keywords:** Kentucky bluegrass, metabolites, rhizome, phytohormone, zeatin

## Abstract

**Simple Summary:**

Rhizomes are underground stems that grow horizontally from the base of the stem and confer competitive, reproductive, foraging, and regenerative advantages to plants. They are especially important for forage and turfgrasses. Different phytohormones influence the growth and development of plant rhizomes in a variety of ways. Here, we compare the phytohormonal and metabolomic profiles of rhizomes from two contrasting Kentucky bluegrass (*Poa pratensis*), a perennial herb with variable rhizome traits materials: multi-rhizome and few-rhizome. We found that zeatin (ZT) was abundant in the multi-rhizome material, while indole-3-acetic acid (IAA), gibberellic acid (GA3), and abscisic acid (ABA) showed the opposite trend. Metabolomics revealed that the ABC transporter pathway and the histidine metabolism pathway, both related to cytokinin, were significantly expressed in the comparison group. Rhizomes are an ideal plant trait. Our findings provide new insights into the development of perennial rhizomatous herbs and a theoretical basis for the future breeding of rhizomatous plants.

**Abstract:**

Rhizome growth and development is regulated by phytohormone. However, endogenous phytohormones affect rhizome initiation, and sustained growth in perennial grass species remains elusive. In this study, we investigated the morphological characteristics and the content of indole-3-acetic acid (IAA), zeatin (ZT), gibberellic acid (GA3), and abscisic acid (ABA) in the rhizomes of two different Kentucky bluegrass. Using ultra-performance liquid chromatography/tandem mass spectrometry (UPLC-MS/MS), we performed metabolite analysis of two different rhizomes. In our study, the multi-rhizome Kentucky bluegrass material ‘Yuzhong’ had an average of 1113 rhizomes, while the few-rhizome material ‘Anding’ had an average of 347 rhizomes. The diameter of rhizome and length of rhizome internode in ‘Yuzhong’ were 1.68-fold and 1.33-fold higher than that of the ‘Anding’, respectively. The rhizome dry weight of ‘Yuzhong’ was 75.06 g, while the ‘Anding’ was 20.79 g. ‘Yuzhong’ had a higher ZT content (5.50 μg·g^−1^), which is 2.4-fold that of ‘Anding’ (2.27 μg·g^−1^). In contrast, the IAA, ABA, and GA3 content of rhizome were markedly higher in ‘Anding’ than ‘Yuzhong’. Correlation analysis revealed significant correlations between ZT and ZT/ABA with the number of rhizomes, diameter of rhizome, and length of rhizome internode, whereas IAA, ABA, GA3, and IAA/ZT were opposite. In the metabolic profiles, we identified 163 differentially expressed metabolites (DEMs) (60 upregulated and 103 downregulated) in positive ion mode and 75 DEMs (36 upregulated and 39 downregulated) in negative ion mode. Histidine metabolism and ABC transporters pathways were the most significantly enriched in the positive and negative ion mode, respectively, both of which are involved in the synthesis and transport of cytokinin. These results indicate that cytokinin is crucial for rhizome development and promotes rhizome germination and growth of Kentucky bluegrass.

## 1. Introduction

Rhizome, a key building block of higher plants, is crucial for plant growth and productivity and has significant biological implications [1]. Rhizome arises from underground stem base meristem to form rhizome buds and horizontal outgrowth [2]. As modified stems, rhizomes possess distinct nodes and internodes, while their leaves have evolved into small, hard scales that protect rhizome growth within the soil [2,3]. The rhizome bud can produce a node that continues the horizontal growth of the rhizome or enter a developmental transition stage where its tip bends upward to generate a nutritional clone of the parent plant [4]. Rhizomes have a powerful storage function that can provide the supply of nutrients for daughter plant growth, regeneration after disturbance, and overwintering rejuvenation, as well as serving as a bridge for physiological integration between ramets [5,6]. Rhizomes are capable of superior traits of mother plant, with important implications for forage and turf grass breeding. The number of rhizomes is determined by the number of rhizome bud formation and effective rhizome bud outgrowth. Phytohormones play important regulatory roles in rhizome growth and development [7].

Auxin, a core phytohormone for plant growth and development, exhibits polar transport [8,9]. Auxin stimulates cell division and differentiation, leading to the production of lateral and adventitious roots. On the other hand, auxin regulates apical dominance [10]. In rice (*Oryza sativa* L.), Xia et al. reported that the auxin response mutant dc1 exhibits enhanced apical dominance and reduced tillering, revealing a role for auxin signaling in shoot branching [11]. Previous research has suggested that exogenous auxin can inhibit tillering in rice [12], wheat (*Triticum aestivum*) [13], and maize (*Zea mays*) [14]. However, auxin cannot directly access axillary buds. Instead, the inhibition of axillary bud growth is primarily due to main stem auxin inhibiting auxin egress from axillary buds [15]. In the second messenger model, axillary bud growth and development are directly regulated by cytokinin (CK) and strigolactone (SL), which act as promoters and inhibitors, respectively [15]. CK is considered to be a promoter of bud growth and development, and it antagonizes the effects of SL on bud growth. CK is transported via the xylem to the lateral buds and stimulates their growth [16]. CK biosynthesis is regulated by local auxin in axillary buds via the phosphoadenosine isopentyl transferase (IPT) gene. Stem node tissues first increase CK biosynthesis then transport CK to dormant buds to stimulate their growth [17]. Qiu et al. found that lateral buds with high auxin activity are dormant, whereas those with high cytokinin activity are active. Removal of the apical bud or exogenous application of cytokinin can break bud dormancy [18]. The exogenous application of CK and gibberellin (GA) promoted rhizome formation and growth, respectively, during the germination and growth of tall fescue (*Festuca arundinacea*) rhizomes [4]. GA also broke the dormancy of rhizomes of *Paris polyphylla* [19]. Tenreira et al. showed that GA induced stolon differentiation in strawberry (*Fragaria vesca*) [20]. However, some studies have suggested that GA promotes SLR 1 protein degradation, leading to reduced tiller number [21]; GA-deficient and overexpressing mutants both exhibited increased branching [22]. GA is particularly important in regulating plant height. The number of branches is generally negatively correlated with plant height. In addition, abscisic acid (ABA) is an inhibitor of branching, and BRC 1 positively regulates ABA synthesis, promoting bud dormancy [22]. However, some studies have indicated that ABA does not directly regulate tillering but rather indirectly modulates SL synthesis, further affecting tillering [23]. Moreover, a dynamic balance between phytohormones is also an important prerequisite for regulating plant growth, such as a high IAA/CK ratio inducing root development, whereas a low ratio induced bud differentiation [24]. Until now, a few phytohormones ratios in the rhizomes remain largely unexplored.

Kentucky bluegrass is a rhizomatous perennial and an important lawn plant widely planted in temperate regions due to its developed rhizome system, aesthetic, and soft leaves [25]. The plant is widely used in water and soil conservation and ecological restoration because it grows rapidly and protects the soil surface from erosion. The rhizomes of Kentucky bluegrass play a crucial role in the reproduction and renewal of populations and plant resistance [26]. Despite their importance, few studies have investigated the rhizomes of forage and lawn grasses, and none of the model plants, such as *Arabidopsis thaliana* or rice, possess rhizomes. The mechanisms underlying the regulation and distribution of endogenous phytohormones in the rhizome of Kentucky bluegrass are still not understood.

Therefore, the goal of the present study was to (1) explore the relationships between different phytohormones and rhizome morphology; (2) identify the key endogenous phytohormones that regulate rhizome germination and growth.

## 2. Results

### 2.1. Morphology Difference of Rhizomes

The rhizomes of Kentucky bluegrass grow vigorously in the spring and fall and become dormant in the summer and winter with the change in growth temperature. During the fall of the second year of the planting, the rhizome phenotypic characteristics of two different Kentucky bluegrass are depicted in Appendix A and Figure 1. The number of rhizomes in ‘Yuzhong’ was significantly higher than that in ‘Anding’ (Figure 1A); the rhizome number of ‘Yuzhong’ was 1113 per plant on average, while the rhizome number of ‘Anding’ was 347 per plant. Subsequently, we assessed the diameter of rhizome and length of rhizome internode in the first-order rhizomes of two materials. The diameter of rhizome and length of rhizome internode in ‘Yuzhong’ were 1.68-fold and 1.33-fold higher than that of the ‘Anding’, respectively (Figure 1B,C). The dry weight of rhizomes in ‘Yuzhong’ was significantly higher than that in ‘Anding’ (Figure 1D).

### 2.2. Endogenous Phytohormone Content of Rhizome

Large differences in the endogenous phytohormone content were observed between the different rhizomes (Figure 2A–D). The ZT content of rhizome was 2.4-fold more abundant in ‘Yuzhong’ than in ‘Anding’ (Figure 2B). In contrast, the IAA, ABA, and GA3 contents of rhizome were markedly higher in ‘Anding’ than ‘Yuzhong’, with ‘Yuzhong’ having only 73.31%, 69.04%, and 75.08% of ‘Anding’, respectively (Figure 2A,C,D).

Endogenous phytohormones can influence each other through synthesis, transport, and metabolism to reach a balanced equilibrium that affects plant growth and development. We therefore analyzed the difference in the phytohormones ratio in different rhizomes. The IAA/ZT ratio was significantly higher in ‘Anding’ than ‘Yuzhong’, whereas the reverse is true for ZT/ABA (Figure 2E,G). There was no significant difference in IAA/GA3 between different rhizomes (Figure 2F). Correlation analysis revealed that the number of rhizome, diameter of rhizome, and length of rhizome internodes were significantly positively correlated with ZT and ZT/ABA, while IAA showed the opposite trend (Figure 2H).

### 2.3. Multivariate Statistical Analysis of Rhizome Metabolomics

To further investigate the difference in the plant signaling pathway of two rhizomes, we used ultra-performance liquid chromatography/tandem mass spectrometry (UPLC-MS/MS) to analyze the expressed metabolites. A total of 1376 metabolites were detected. Principle component analysis (PCA) showed the separation of the samples into two groups, with the first principal component (PC1) and the second principal component (PC2) accounting for 51.08% of the total variation (Figure 3A). This suggested that metabolite profiles had concordance with rhizome morphology. The clustering heatmap showed that each sample was clustered into its own category, with significant differences in metabolites between groups (Figure 3B). 

### 2.4. Differentially Metabolites

Partial least-squares discriminant analysis (PLS-DA) was used for metabolic profiling analysis (Figure 4). The scatter plot showed significant separation and the model was stable and feasible (R2Y and Q2 are close to 1, and R2Y > Q2). PC1 and PC2 accounted for 28.80% and 15.89% of the total variation in the positive ion mode (R2Y = 0.98, Q2Y = 0.84) and 37.08% and 16.75% in the positive ion mode (R2Y = 0.99, Q2Y = 0.94). PLS-DA was a supervised multivariate regression method that was used to model the relationship between the information on the metabolites and the different rhizomes. The threshold value was set to VIP > 1.0, FC > 2 or FC < 0.5, and *p* < 0.05. In positive ion mode, 163 differentially expressed metabolites (DEMs) were identified, of which 60 DEMs were upregulated and 103 DEMs were downregulated; in negative ion mode, 75 DEMs were identified, of which 36 DEMs were upregulated and 39 DEMs were downregulated (Table 1).

### 2.5. KEGG Pathway

Pathway analysis of the DEMs was performed based on the KEGG database. Histidine metabolism and ABC transporters were screened for significant enrichment in positive and negative ion mode, respectively. Additionally, plant hormone signaling DEMs were also enriched in positive ion mode (Figure 5). Nine DEMs were selected based on their association with these three KEGG pathways (Table 2).

### 2.6. Interaction of Metabolites and Phytohormones

Large differences in key metabolites of the histidine metabolism pathway and the ABC transporters pathway were observed between the two rhizomes (Figure 6). Imidazoleacetic acid, alpha-ketoglutaric acid, and 3-methylhistidine, which are involved in histidine catabolism, were significantly higher in ‘Yuzhong’ than in ‘Anding’. These results suggest that ‘Yuzhong’ has a higher rate of histidine breakdown than ‘Anding’, which may have implications for their physiological functions. The opposite trend was observed in the ABC metabolic pathway (L-Serine, Inositol, Raffinose).

The correlation between phytohormones and metabolites results is shown in Figure 7. There was a positive correlation between ZT and ZT/ABA and L-histidine, 3-methylhistidine, alpha-ketoglutaric acid, and salicylic acid, with alpha-ketoglutaric acid and salicylic acid being significantly positively correlated, and 3-methylhistidine being highly significantly positively correlated. Raffinose, L-serine, inositol, and urocanic acid were positively correlated with IAA, GA3, ABA, IAA/ZT, and IAA/GA3, where raffinose and inositol were significantly positively correlated with IAA, ABA, and IAA/ZT.

## 3. Discussion

Plants exhibit phenotypic plasticity in response to habitat heterogeneity by altering their morphological characteristics and cellular metabolism to form stable heritable structural features [27]. Under the influence of endogenous phytohormones and environmental signals, plants display general heterogeneity in branching development patterns, growth rates, and other traits throughout their life cycle [28]. This phenotypic plasticity enables plants to evolve in response to harsh environments by altering their morphology and physiology to enhance their survival and reproduction [29,30]. We compared the rhizomes of two wild Kentucky bluegrass from different regions and found phenotypic differences in rhizome number, rhizome diameter, length of rhizome internode, and dry weight of rhizome that may reflect differences in their original habitat (Figure 1 and Appendix A). Rhizomes are the main mode of asexual reproduction and population expansion in Kentucky bluegrass. Strong rhizomatous plants can expand horizontally across different resource patches and access and use local resources through source–sink relationships between parent and offspring plants, showing that plant competition, reproduction, and nutrient acquisition are not limited by vertical space, seeds, or roots [5,6].

Rhizome growth and development is a complex process regulated by genetic and environmental factors, where phytohormone regulation may be a conserved mechanism in the germination and growth of rhizome [16]. Exogenous phytohormones can alter endogenous phytohormone levels and thereby regulate rhizome development [4]. CK stimulates bud formation, and receptor genes such as IPT, a CK synthesis gene, and histidine kinase both increase with CK levels [4]. CK biosynthesis is first increased in stem node tissue, and then CK is transported to dormant buds to stimulate their growth. Previous investigations by Ma et al. demonstrated that rhizome bud formation is activated by CK in the tall fescue [4]. Our study also found high ZT contents in multi-rhizomes Kentucky bluegrass (Figure 2). Axillary bud growth is regulated by the effect of IAA on ZT rather than by the absolute level of individual phytohormones. IAA antagonizes ZT by suppressing its local synthesis and translocation, preventing ZT from entering the axillary buds and inhibiting their growth [31]. The strength of apical dominance is determined by IAA/ZT, and the exogenous application of NAA (naphthaleneacetic acid, a synthetic auxin) significantly reduces ZT levels and leads to fewer tiller buds [32]. Kentucky bluegrass is a rhizomatous caespitose plant with both tillers and rhizomes; these derive from homologous meristems and may share common regulatory mechanisms [33]. Histidine receptor kinase can sense and bind cytokinin, followed by the autophosphorylation of histidine; treatment with 6-benzylaminopurine (a synthetic cytokinin) was able to stimulate rhizome formation and increase the levels of the histidine kinases HK1 and cytokine signaling pathway receptor [4]. Of the differentially expressed metabolites we detected, most of the metabolites in the histidine metabolic pathway were significantly higher in ‘Yuzhong’ than in ‘Anding’ (Figure 6). Some studies have reported that increasing the supply of the histidine synthesis pathway can promote the growth and development of rhizomes and vice versa [34,35]. Increased CK content may upregulate histidine receptor kinase and its regulatory homologs RR1 and RR6 [4]. ABC transporters participate in cytokinin transportation through the xylem and influence the vascular bundle development [36]. We also found that ABC transporters were significantly enriched in the negative ion mode (Figure 5B). Environmental signals may cause the directional regulation of downstream transcriptional targets by cytokinin [16]. The original habitats showed stronger differences between the ‘Yuzhong’ and ‘Anding’. Though long-term natural selection and evolution, it produces a small number of rhizomes and ramets, which reduces the nutrient transfer from mother plant to offspring, forming a growth mode more suitable for individual reproduction and population propagation [37]. ABC transporters transport hormones, resulting in hormone inactivity or reduced intracellular hormone activity [38]. In our study, ABC transporter-related metabolites were higher in ‘Anding’ than in ‘Yuzhong’.

Auxin is one of the most important phytohormones which affect plant morphogenesis and development through synthesis, transport, and signal transduction pathways. In the present study, ‘Yuzhong’ had a higher rhizome number, rhizome diameter, and internode length than ‘Anding’, while the opposite trend was observed for the IAA content. Thimann et al. [39] showed that increasing the IAA polar transport flow in the main stem of broad bean (*Vicia faba*) inhibited the growth and development of tiller buds. Each phytohormone accumulates and acts in response to the interactions with other phytohormones, genetics, and environment [13]. Few-rhizome material with high GA3 and ABA levels inhibited the emergence of rhizome in this study (Figure 2). GA3 is capable of suppressing the growth of tillers in crops [40], possibly by increasing the IAA content and IAA/ZT ratio and decreasing ZT levels in tiller nodes [13]. However, other studies have demonstrated that the GA inhibition of tillering is not related to IAA but rather involves the upregulation of cytokinin degradation genes to reduce CK levels [41]. GA has also been shown to activate rhizome sprouting in *Paris polyphylla* [19] and to induce stolon differentiation in strawberry [20]. These effects are likely due to the species-specific mechanism of GA action on branching [24]. While GA3 generally promotes stem elongation [21], our study found that the internode length of rhizomes in ‘Anding’ was significantly shorter than that of ‘Yuzhong’. This may be because GA3 had a greater effect on rhizome number than on internode elongation in *Poa pratensis*. Additionally, GA3 may also influence ABA levels, which are closely related to lateral shoot dormancy; increasing ZT/ABA can weaken apical dominance [42], and the addition of exogenous ABA can inhibit the growth of wheat tillering [31]. Some investigators consider ABA to not act as a messenger in bud growth; instead, it mainly regulates the bud dormancy process by responding to environmental factors [40]. Rhizomes are horizontal underground stems that are minimally affected by visible light. Here we showed that the DEMs associated with jasmonic acid were enriched in plant signaling pathways (Figure 5 and Table 2). This finding is consistent with previous reports on the role of jasmonic acid in the rhizome of *Oryza longistaminat* [33]. However, we greatly regret that this study failed to identify the jasmonic acid content, which limits our understanding of its mechanisms in rhizome.

## 4. Materials and Methods

### 4.1. Plant Materials and Growth Conditions

Two wild Kentucky bluegrass with different rhizomes were used in this study (Appendix A). Multi-rhizome material (Yuzhong) and few-rhizome material (Anding) were collected from Yuzhong County (35°48′ N, 104°04′ E; altitude 1965 m) and Anding County (35°58′ N, 104°62′ E; altitude 2035 m), Gansu province, China, respectively. The seeds were stored in the seed storage room of Grassland and science college, Gansu agricultural university. An experiment was performed from 15 June 2019 to 15 October 2020 in the experimental field of Gansu agricultural university (Lanzhou, China; 36°48′ N, 103°3′ E, Altitude 1517 m). On 15 June 2019, each material was planted in 15 plots randomly arranged in the field. After emergence, the seedlings were thinned down to one plant with a similar size per plot. The spacing between the plants was 70 cm. The plants were watered, fertilized, and otherwise normally managed (Appendix A).

### 4.2. Determination of Rhizome Morphology

On 15 October 2020, all the plants were carefully removed from the soil, and the 1st-order rhizomes were separated. Each plant was a replicate. Then, three plants were randomly selected for the measurement of rhizome morphological parameters, including rhizome number, rhizome diameter, and rhizome internode. The rhizome dry weight was determined after drying for 48 h at 80 °C. The remaining were stored at −80 °C in an ultra-low temperature freezer (Thermo Scientific, MA, USA).

### 4.3. Determination of Phytohormone

Endogenous IAA, ZT, GA3, and ABA were determined by the method outlined by Yang et al. [43] with some modification. Rhizome samples (2 g) were ground in liquid nitrogen and extracted with 80% cold methanol. The homogenate was incubated at 4 °C for 16 h and centrifuged at 10,000 g for 15 min at 4 °C. The supernatant was collected, and the residue was extracted with 80% cold methanol using an ultrasonicator bath (KQ-700VDV, Shanghai, China) for 30 min. This procedure was repeated three times and the supernatants were pooled. The pooled supernatant was concentrated using a vacuum centrifugal concentrator (RVC 2-25 CD plus, Christ, Osterode am Harz, Germany) at 40 °C for 4 h and extracted with petroleum ether. After centrifugation, the upper ether phase was discarded, and the aqueous solution was extracted with ethyl acetate three times. Phytohormone extract was concentrated in a vacuum centrifugal concentrator, and we finalized the volume to 2 mL with 80% cold methanol. The extract was filtered through a 0.22 µm membrane filter and analyzed via high-performance liquid chromatography (HPLC, ACQURITY Arc, MA, USA). The mobile phases were methanol (mobile phase A) and 0.1% phosphoric acid (mobile phase B). The flow rate was 1.0 mL·min-1 and the detection wavelength was 254 nm (Waters Symmetry c-18, Milford, MA, USA). Three biological replicates were performed for each material.

The quantification was performed using the external standard method. The standards ZT (0.0013 g), GA3 (0.0207 g), IAA (0.0012 g), and ABA (0.0010 g) were accurately weighed and then diluted to 5 mL with chromatographically pure methanol to make the standard stock solution with concentrations of 260 μg·m L^−1^, 4140 μg·m L^−1^, 240 μg·m L^−1^, and 200 μg·m L^−1^, respectively. The standard stock solution was diluted to a range of concentrations, including single-label and mixed-label for quantification (due to the low response value of GA3 at 254 nm, the content of GA3 in the standard solution was increased to 20 times that of the other hormones after several measurements). The linear regression equations and correlation coefficients of the endogenous hormones were calculated by sequentially injecting the series of concentration standards into the sample under the above chromatographic conditions, using the concentration (X) as the horizontal coordinate and the peak area (Y) as the vertical coordinate (Table 3).

### 4.4. Metabolite Extraction

The frozen rhizomes were used for metabolomics analysis. Six biological replicates were performed for each material. In total, 100 mg of rhizomes was ground in liquid nitrogen. Then, 500 μL of 80% methanol solution was added and allowed to stand for 5 min. The homogenate was centrifuged at 15,000× *g* for 20 min at 4 °C. The supernatant was collected, and the extraction was repeated. Then, 10 μL of each sample was removed and pooled as a quality control sample [44].

### 4.5. Ultra-Performance Liquid Chromatography/Tandem Mass Spectrometry (UPLC-MS/MS) Analysis

UPLC-MS/MS analysis was performed using a Vanquish UHPLC system (Thermo Fisher Scientific, Darmstadt, Germany). The mobile phase consisted of water containing 0.1% formic acid (phase A) and methanol (phase B) in positive polarity mode and 5 mM ammonium acetate (pH 9.0; phase A) and methanol (phase B) in negative polarity mode. The flow rate was 0.2 mL/min and the column temperature was 40 °C. Mass spectrometric conditions included a spray voltage of 3.2kV, sheath gas flow rate of 40 arb, aux gas flow rate of 10 arb, and capillary temperature of 320 °C. The scan range was from *m*/*z* 100 to 1500.

Identification of metabolites was performed using the novoDB database. Data and chromatogram peak were analyzed by SCIEX OSV1.4 (Ontario, Canada). Metabolites were annotated using the KEGG database (https://www.genome.jp/kegg/pathway.html, accessed on 2 May 2022), HMDB database (https://hmdb.ca/metabolites, accessed on 2 May 2022) and LIPID Maps database (http://www.lipidmaps.org/, accessed on 2 May 2022).

Multivariate statistical analyses were performed using the metabolomics data processing software metaX (http://metax.genomics.cn/, accessed on 2 May 2022) to transform the data into principal component analysis (PCA) and partial least-squares discriminant analysis (PLS-DA) to obtain the VIP value of each metabolite. In the univariate analysis section, the statistical significance (*p*-value) of each metabolite between the two groups was calculated based on the *t*-test, and the fold change in metabolites between the two groups was calculated as the FC value. The standard criteria for differential metabolite screening were VIP > 1, *p*-value < 0.05, and FC ≥ 2 or FC ≤ 0.5. Correlation analysis between the different metabolites was performed using the R cor, and statistical significance was realized using cor.mtest in R. A *p*-value < 0.05 was considered statistically significant, and correlation plots were drawn using the R corrplot package. Bubble plots were generated using the ggplot2 package in software from R Studio (R version R-3.4.3) and the KEGG database was used to investigate the functions and metabolic pathways of the metabolites. KEGG pathways significantly enriched in differential metabolites were screened using a threshold of *p* ≤ 0.05.

### 4.6. Statistical Analysis

Data were analyzed using SPSS version 25.0 (SPSS, Inc., Chicago, IL, USA). The difference between the two materials in all indicators was evaluated by Student’s *t*-test. Results are depicted as means with standard errors. Plots for the figure were produced using OriginLab.

## 5. Conclusions

In the present study, we investigated the role of endogenous phytohormones and metabolism in the growth and development of rhizomes in multi-rhizome and few-rhizome Kentucky bluegrass, demonstrating that phytohormones are important materials in rhizome. Among them, we found that ZT was abundant in the multi-rhizome Kentucky bluegrass. ZT positively correlates with the rhizome number, diameter, internode length, and dry weight of rhizome. There were high concentrations of IAA, ABA, and GA3 in the few-rhizome Kentucky bluegrass. Additionally, histidine metabolism pathways and ABC transporter pathways related to cytokinin were significantly enriched. This finding may suggest that ZT plays a crucial role in regulating rhizome growth and development, and that IAA, ABA, and GA3 may inhibit this process. In addition, ZT may act through histidine metabolic pathways and ABC transporter pathways to affect rhizome growth and development. This provides a scientific basis for the application of rhizome in the rhizomatous plants. Future research should focus on establishing a comprehensive network regulating rhizome growth and developing new cultivars with ideal plant types through molecular breeding to improve the application of rhizomes in production.

## Figures and Tables

**Figure 1 biology-12-01120-f001:**
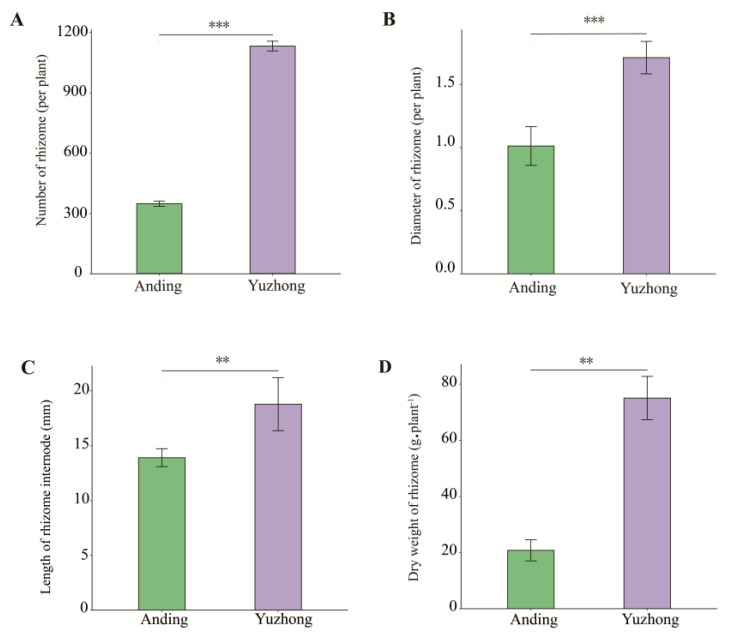
Morphological differences between the rhizomes of two materials. (**A**) Number of rhizomes. (**B**) Diameter of rhizome. (**C**) Length of rhizome internode. (**D**) Dry weight of rhizome. Significance of differences was based on Student’s *t*-test. Asterisks means significant differences between two materials (** *p* < 0.01, *** *p* < 0.001).

**Figure 2 biology-12-01120-f002:**
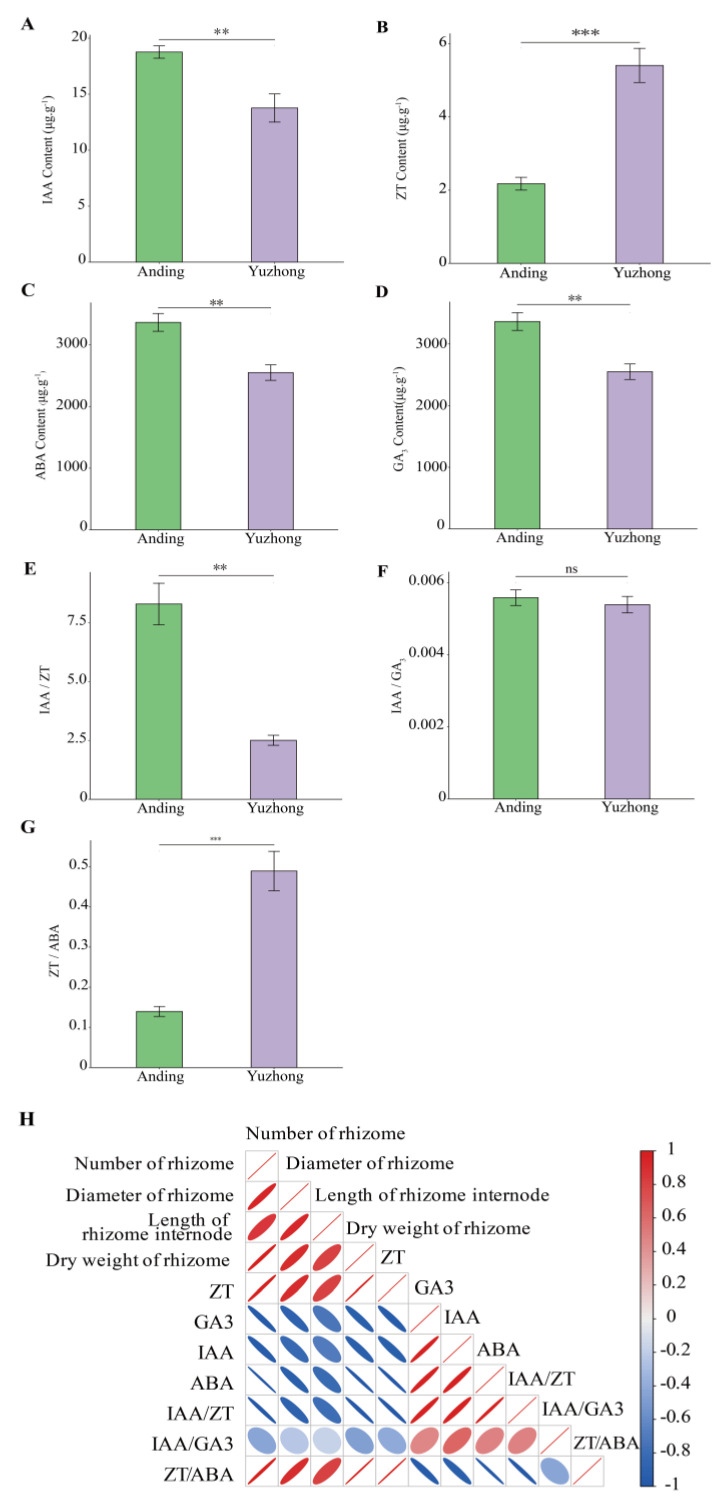
(**A**–**D**) Differences in endogenous phytohormones content in the rhizomes of two materials. (**A**) Indole-3-acetic acid (IAA), (**B**) zeatin (ZT), (**C**) abscisic acid (ABA), and (**D**) gibberellic acid (GA3) content in two different rhizomes. (**E**–**G**) Phytohormone ratios in the rhizomes of two materials. (**E**) IAA to ZT ratio, (**F**) IAA to GA3, and (**G**) ZT to ABA ratio of two different rhizomes. (**H**) Correlation analysis between rhizome phenotype and phytohormone. The red represents positive correlation, and blue represents negative correlation. Significance of differences was based on Student’s *t*-test. Asterisks means significant differences between two materials (** *p* < 0.01, *** *p* < 0.001).

**Figure 3 biology-12-01120-f003:**
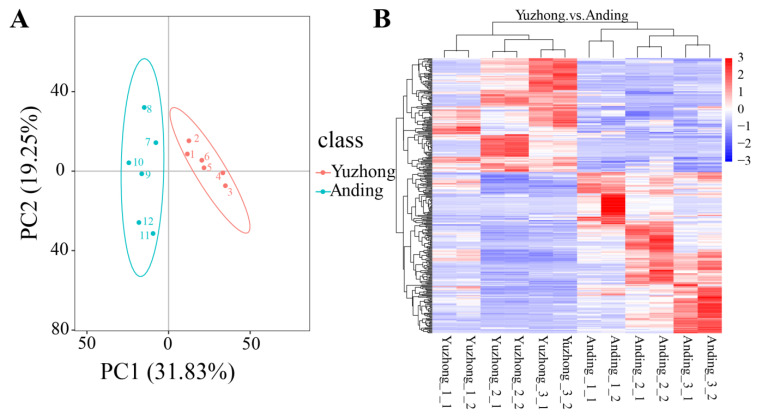
(**A**) Principal component analysis of rhizome metabolites. Each dot represents a sample from independent biological replicates. (**B**) Cluster heatmap of differential metabolites. The horizontal axis is clustering of metabolites, and the vertical axis for grouping of samples. Each sample has six biological replicates.

**Figure 4 biology-12-01120-f004:**
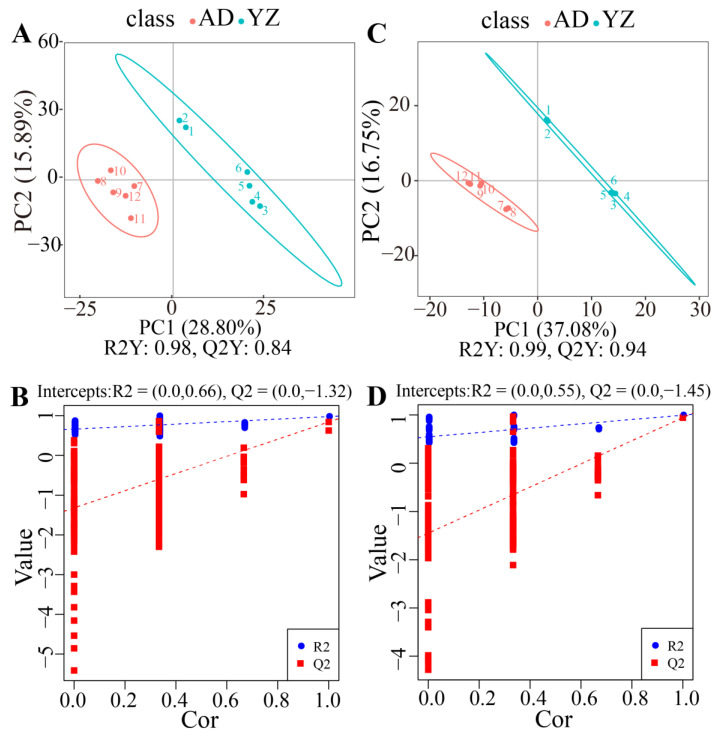
(**A**) Scatter plot of PLS-DA in positive ion mode, (**B**) sort test map of PLS-DA in positive ion mode, (**C**) and scatter plot of PLS-DA in negative ion mode. The horizontal axis is the score of the sample on the first principal component; the vertical axis is the score of the sample on the second principal component. (**D**) Sort test map of PLS-DA in negative ion mode. The horizontal axis is the correlation between the random group and original group; the vertical axis is the score of R2 and Q2.

**Figure 5 biology-12-01120-f005:**
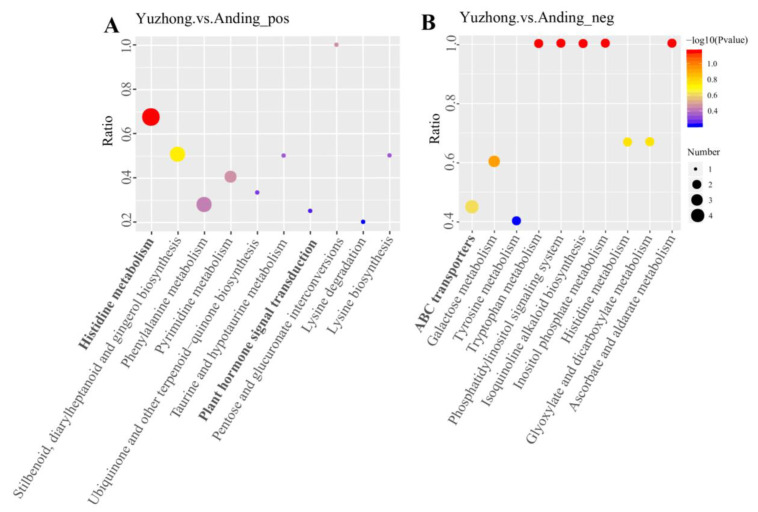
(**A**) KEGG bubble map of differentially enriched metabolites in positive ion mode; (**B**) KEGG bubble map of differentially enriched metabolites in negative ion mode.

**Figure 6 biology-12-01120-f006:**
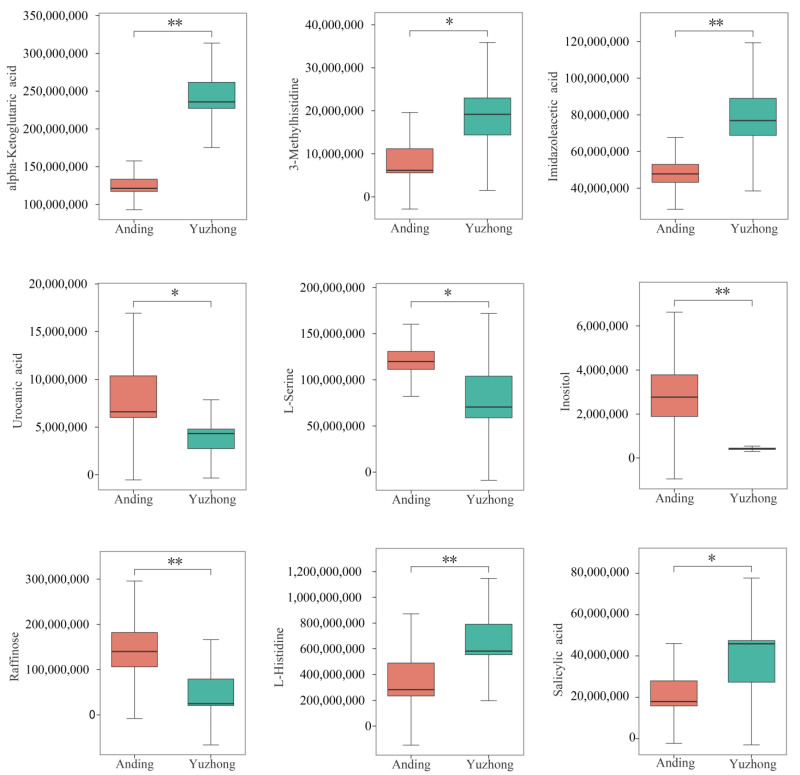
Content of key differentially expressed metabolites. Asterisks means significant differences between two materials (* *p* < 0.05, ** *p* < 0.01).

**Figure 7 biology-12-01120-f007:**
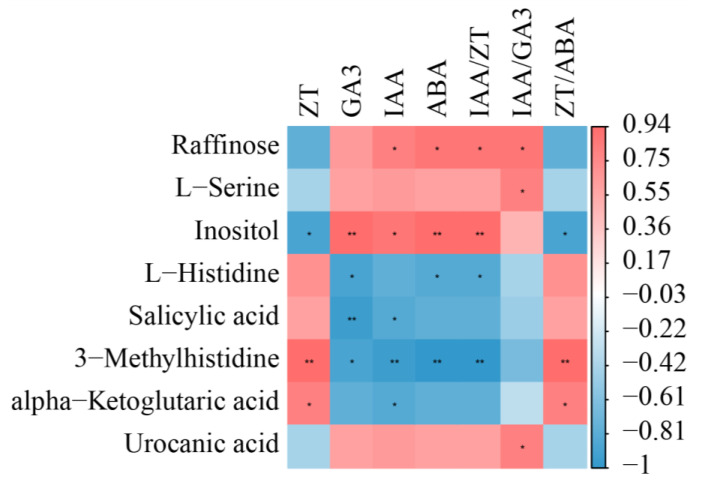
Correlation analysis between rhizome phytohormone and metabolites. Red represents positive correlation, and blue represents negative correlation. Asterisks means significant differences between two materials (* *p* < 0.05, ** *p* < 0.01).

**Table 1 biology-12-01120-t001:** Differentially expressed metabolites (DEMs) in the positive/negative ion mode.

	Total Differentially Expressed Metabolites (DEMs)	Up DEMs	Down DEMs
Positive ion mode	163	60	103
Negative ion mode	75	36	39

**Table 2 biology-12-01120-t002:** Changes in differentially expressed metabolites in comparison group.

Metabolite Name	KEGG Pathway	VIP	log_2_FC	*p*-Value
Imidazoleacetic acid	Histidine metabolism	1.0594	0.7486	0.0304
3-Methylhistidine	Histidine metabolism	1.7679	1.4103	0.0027
Alpha-Ketoglutaric acid	Histidine metabolism	1.4028	0.9658	0.0003
Urocanic acid	Histidine metabolism	1.0687	−1.1618	0.0447
Raffinose	ABC transporters	1.6006	−1.6531	0.0060
L-Serine	ABC transporters	1.3568	−0.6028	0.0251
Inositol	ABC transporters	1.8831	−2.6496	0.0000
L-Histidine	ABC transporters	1.4208	0.9078	0.0109
Salicylic acid	Salicylic acid	1.0364	0.8770	0.0448

**Table 3 biology-12-01120-t003:** Peak times and linear equations of four hormones.

Phytohormone	Peak Time (min)	Regression Equation	R2
IAA	4.280	Y = 1.66 × 104X − 1.98 × 103	0.999943
ZT	6.030	Y = 4.36 × 10X + 1.34 × 103	0.999464
GA3	7.987	Y = 4.58 × 103X − 2.94 × 102	0.999656
ABA	8.794	Y = 2.60 × 104X − 2.70 × 103	0.999934

## Data Availability

The data presented in this study are available in the article and the Appendix A here.

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
