# Peer review of "Cytokinin and Metabolites Affect Rhizome Growth and Development in Kentucky Bluegrass (Poa pratensis)"

_biology, 2023, doi:10.3390/biology12081120_

Round 1

Reviewer 1 Report

General comments:

The authors investigated the morphological characteristics and the content of indole-3-acetic acid (IAA), zeatin (ZT), gibberellic acid (GA3), and abscisic acid (ABA) in the rhizomes of two different Poa pratensis. Using ultra-performance liquid chromatography/tandem mass spectrometry (UPLC-MS/MS), they performed metabolite analysis of two different rhizomes. The results revealed 163 differentially expressed metabolites (DEMs) (60 upregulated and 103 downregulated) in positive ion mode and 75 DEMs (36 upregulated and 39 downregulated) in negative ion mode. The manuscript holds scientific potential but before publication, some suggestions are recommended.

The manuscript can be accepted after minor revisions. A moderate English editing and spell check is required.  A few typographical errors also need to be addressed.

Comments:

L83: Write the common name of Poa pratensis.

Introduction: In the introduction section, the authors should highlight the importance of the present study with supporting literature. The introduction seems more like a review.

In Figure 1: what is the control?

Discussion: Can be improved with the latest references supporting the findings of the present study.

There is no section supporting the protein expression studies. It should be added.

Materials and methods:

Which extraction methods have been used for protein extraction. It is not mentioned.

Conclusions: It should be well written and highlight the present study's significant findings.

Moderate English editing and spell check is required. 

Author Response

Dear editor,

We would like to thank you for taking the time to read our manuscript and giving us the opportunity to revise this manuscript. We honestly appreciated the comments, which highlighted the thorough attention to the content, and we tried our best to fulfil them. We have studied all these comments carefully and made corresponding corrections. We respond these comments point by point as below. We hope the new manuscript will meet your magazine’s standard. If any questions still remain, please let us know. We are looking forward to hearing good news from you! Thanks for your attention and patience!

Yours sincerely,

Xiaoming Bai

Comment

The authors investigated the morphological characteristics and the content of indole-3-acetic acid (IAA), zeatin (ZT), gibberellic acid (GA3), and abscisic acid (ABA) in the rhizomes of two different Poa pratensis. Using ultra-performance liquid chromatography/tandem mass spectrometry (UPLC-MS/MS), they performed metabolite analysis of two different rhizomes. The results revealed 163 differentially expressed metabolites (DEMs) (60 upregulated and 103 downregulated) in positive ion mode and 75 DEMs (36 upregulated and 39 downregulated) in negative ion mode. The manuscript holds scientific potential but before publication, some suggestions are recommended.

1. The manuscript can be accepted after minor revisions. A moderate English editing and spell check is required. A few typographical errors also need to be addressed.

Response: Thank you very much for this comment. The manuscript has been checked sentence by sentence, and spelling and grammar mistakes have been corrected. We have formatted all the typographical in the manuscript in line with journal requirements. Thanks again for your comments.

2. L83: Write the common name of Poa pratensis.

Response: Thank you very much for this comment. In our revised manuscript, we have added the common name (Kentucky bluegrass) of Poa pratensis.

3. Introduction: In the introduction section, the authors should highlight the importance of the present study with supporting literature. The introduction seems more like a review.

Response: Thank you very much for this comment. We have carefully revised the section according to your suggestions. (page 2). Thanks again for your comments.

4. In Figure 1: what is the control?

Response: Thank you very much for this comment. In this study, we have analyzed the t-test for the different indicators of the two materials. So, we're just comparing two materials to each other.

5. Discussion: Can be improved with the latest references supporting the findings of the present study.

Response: Thank you very much for this comment. We have revised this part as your suggestions.

6. There is no section supporting the protein expression studies. It should be added.

Response: Thank you very much for this comment. This suggestion is valuable for a comprehensive study of phytohormone in rhizomes. However, the objective of this paper is to study the metabolism of phytohormone in rhizomes, mainly explaining the content and role of phytohormone and metabolite compounds in the rhizome. Your comments are important and we will take it to heart. Therefore, we intend to continue this line of inquiry on rhizome in a follow-up paper. Once again, thank you very much for your comments and suggestions.

7. Which extraction methods have been used for protein extraction. It is not mentioned.

Response: Thank you very much for your very constructive comments. In our upcoming study, we will conduct the rhizome proteomics based on your suggestion. We will provide a comprehensive description of the methodology.

8. Conclusions: It should be well written and highlight the present study's significant findings.

Response: Thank you very much for your suggestion. We have re-written this part according to the suggestion in the revised manuscript. (page 13-14 ). Thanks again for your comments.

Reviewer 2 Report

Dear Editor in Chief of biology journal

I studied the article entitled “Differences in endogenous phytohormones and metabolic profiles in different rhizomatous of Poa pratensis” written by Xiaoming et al. this article was written in good format. However, there is a little information into MS so it is necessary to improve the quality and quantity of the article by adding some wanted analysis. The acceptance of this article is recommended after major revision.

introduction part:

please summarize line 30 to 53 in one paragraph.

add one paragraph with diagram about interaction of phytohormones and effect on plant growth such as root and shoot.

line57: Oryza sativa L. is true.

results part:

·         figure 1 is not clear. please provide another with good resolution.

·         figure 2, 3 and 4 are not clear. please provide anothers with good resolution and magnification.

·         please add information about two ecotypes of Poa pratensis in one table such as plant height, shoot yield, number of days to flowering, number of tiller and so on.

·         in figure 2: please add correlation of hormones with other characteristics of poa such as shoot yield, plant height and so on.

·         please add one part in result with title: interaction of metabolites and hormones with cellular components and provide the figure of these interactions by pathway studio software and explain in two paragraph.

·         what differences are there between two ecotypes with view point of metabolites? please add one paragraph in this matter and explain the significant of metabolites by t-test.

·          why there is not any relationship among metabolites and plant signaling hormone in KEGG pathway?

material and methods:

·         line 315: data analysis and plotting were with R software? which analysis? and which plots?

·         please add pictures of two Poa pratensis ecotypes in the pot or farm with good quality.

·         please add other information that mentioned in the result in this part.

Discussion part:

·         please add two paragraph about interaction of hormones, metabolites and related genes.

·         this part also could be summarizing by deleting one of paragraphs.

references part:

References are not in one format. and some of them is related to wheat, barley, animal that there are not any rhizomes.

best wishes

Author Response

Dear editor,

We would like to thank you for taking the time to read our manuscript and giving us the opportunity to revise this manuscript. We honestly appreciated the comments, which highlighted the thorough attention to the content, and we tried our best to fulfil them. We have studied all these comments carefully and made corresponding corrections. We respond these comments point by point as below. We hope the new manuscript will meet your magazine’s standard. If any questions still remain, please let us know. We are looking forward to hearing good news from you! Thanks for your attention and patience!

Yours sincerely,

Xiaoming Bai

Comment

I studied the article entitled “Differences in endogenous phytohormones and metabolic profiles in different rhizomatous of Poa pratensis” written by Xiaoming et al. this article was written in good format. However, there is a little information into MS so it is necessary to improve the quality and quantity of the article by adding some wanted analysis. The acceptance of this article is recommended after major revision.

  1. Please summarize line 30 to 53 in one paragraph.

Response: Thank you very much for your suggestion. We have reorganized the section in the revised manuscript. (page 2).

  1. Add one paragraph with diagram about interaction of phytohormones and effect on plant growth such as root and shoot.

Response: Thank you very much for your suggestion. We have made appropriate additions to this part of the content in the revision manuscript. (page 2).

  1. Line57: Oryza sativa L. is true.

Response: Thank you very much for your suggestion. We have revised it in the revision manuscript.

  1. Figure 1 is not clear. please provide another with good resolution.

Response: Thank you for your good suggestion. We have improved the quality of picture in our revised manuscript.

  1. Figure 2, 3 and 4 are not clear. please provide anothers with good resolution and magnification.

Response: Thank you for your good suggestion. We have improved the quality of picture in our revised manuscript.

  1. Please add information about two ecotypes of Poa pratensis in one table such as plant height, shoot yield, number of days to flowering, number of tiller and so on.

Response: Thank you for your good suggestion. In our study, we focused on the rhizome of Kentucky bluegrass, which is mainly used in lawns. We also added the rhizome dry weight in Figure 1 D. The flowering period, the plant height as a secondary factor, which we have not yet dealt with in this part of the experiment. Therefore, we intend to continue this line of inquiry on Kentucky bluegrass in a follow-up paper. Once again, thank you very much for your comments and suggestions.

  1. In figure 2: please add correlation of hormones with other characteristics of poa such as shoot yield, plant height and so on.

Response: Thank you for your good suggestion. We've already added content in the revised manuscript.

  1. Please add one part in result with title: interaction of metabolites and hormones with cellular components and provide the figure of these interactions by pathway studio software and explain in two paragraph.

Response: Thank you for your good suggestion. We've already added content in the revised manuscript. (page 8-10).

  1. what differences are there between two ecotypes with view point of metabolites? please add one paragraph in this matter and explain the significant of metabolites by t-test.

Response: Thank you for your good suggestion. We've already added content in the revised manuscript. (page 8-10).

  1. Why there is not any relationship among metabolites and plant signaling hormone in KEGG pathway?

Response: Thank you for your good comment. We have carefully checked again the data. In our study, metabolites were also present in plant signaling pathways, but they all showed insignificant differences under rigorous screening. In contrast, the differences were significant in plant hormone synthesis and transport pathways. This leads to the following situation.

  1. line 315: data analysis and plotting were with R software? which analysis? and which plots?

Response: Thank you for your good suggestion. We've already added content in the revised manuscript. (page 13).

  1. Please add pictures of two Poa pratensis ecotypes in the pot or farm with good quality.

Response: Thank you for your good suggestion. The photographs are in supplementary material Figure S2.

  1. Please add other information that mentioned in the result in this part.

Response: Thank you for your good suggestion. Relevant contents have been added as required. (page 12).

  1. Please add two paragraph about interaction of hormones, metabolites and related genes.

Response: Thank you for your good suggestion. Relevant contents have been added as required. (page 10-11).

  1. This part also could be summarizing by deleting one of paragraphs.

Response: Thank you for your good suggestion. We have revised and reorganized this section.

  1. References are not in one format. and some of them is related to wheat, barley, animal that there are not any rhizomes.

Response: Thank you for your good suggestion. We have adjusted the overall reference formatting. Relatively little research has been done on rhizomes of grasses. We have tried our best to revise some of the references. We hope that these changes we have made meet your approval.

Reviewer 3 Report

The study highlights the role of endogenous phytohormone responsible for promoting rhizome initiation and growth in Poa. The comparative analysis of only two rhizomes is a major limitation in this study which require a higher number of samples. Otherwise, the information is valuable and constructive and there are few suggestions for further improvement.

What was the basis for selecting two rhizomes for the analysis?

How will this study be helpful in providing a practical solution for a good rhizome harvest?

Please modify the title for a better representation of the experiments carried out in this study or the major findings

Abstract: Line 8-10: this sentence is incomplete. In fact, the abstract should be modified for highlighting significant findings (statistical data) and future implications of this study.

What is the relevance of lines 29-53 for this study.

Section 2.1, please modify the legend.

latest studies on similar aspects are not included (the year 2022-2023)

Author Response

Dear editor,

We would like to thank you for taking the time to read our manuscript and giving us the opportunity to revise this manuscript. We honestly appreciated the comments, which highlighted the thorough attention to the content, and we tried our best to fulfil them. We have studied all these comments carefully and made corresponding corrections. We respond these comments point by point as below. We hope the new manuscript will meet your magazine’s standard. If any questions still remain, please let us know. We are looking forward to hearing good news from you! Thanks for your attention and patience!

Yours sincerely,

Xiaoming Bai

Comment

  1. The study highlights the role of endogenous phytohormone responsible for promoting rhizome initiation and growth in Poa. The comparative analysis of only two rhizomes is a major limitation in this study which require a higher number of samples. Otherwise, the information is valuable and constructive and there are few suggestions for further improvement.

Response: Thank you very much for your comment. In our previous work, based on rhizome plant number, rhizome volume, rhizome surface area, aboveground plant volume, stem length, tiller number, and largest rhizome extension distance that had the different expansion capacity wild Kentucky bluegrass materials were screened out [1]. Our current study evaluated phytohormones and metabolomes based on a pre-screening of the two most differentiated rhizomes of Kentucky bluegrass.

[1] Chen R, Bai X, Zhen W, Guo X, Huang Q, Li C, Lu N. Study on rhizomes extensibility of 11wild kentucky bluegrass collected in Gansu province. Acta Agrestia Sinica. 2019;27:1250–8.

  1. What was the basis for selecting two rhizomes for the analysis?

Response: Thank you very much for your comment. We selected the Kentucky bluegrass with the greatest differences in rhizome number for further analysis on the basis of our previous work.

  1. How will this study be helpful in providing a practical solution for a good rhizome harvest?

Response: Thank you very much for your comment. Our results showed that CK content was significantly higher in multi-rhizome material than in few-rhizome material, which may indicate that CK is a key hormone in regulating rhizome growth, and therefore, for improvement in contemporary traits, rhizome growth can be promoted by spraying hormone analogs such as CK. In the future breeding of rhizomatous plants, our study provides basic data for molecular breeding.

  1. Abstract: Line 8-10: this sentence is incomplete. In fact, the abstract should be modified for highlighting significant findings (statistical data) and future implications of this study.

Response: Thank you very much for your suggestion. We have revised this part according to your comment. (please see line 20-40).

  1. What is the relevance of lines 29-53 for this study.

Response: Thank you very much for your comment. We have carefully revised the section according to your suggestions. (please see line 44-58). Thanks again for your comments.

  1. Section 2.1, please modify the legend.

Response: Thank you very much for your suggestion. All figures resolution and quality have been improved. Please see them in our revised manuscript.

  1. Latest studies on similar aspects are not included (the year 2022-2023)

Response: Thank you for your good suggestion. We have adjusted the overall reference formatting. Relatively little research has been done on rhizomes of grasses. We have tried our best to revise some of the references. We hope that these changes we have made meet your approval.

Reviewer 4 Report

The manuscript authors entitled “Differences in endogenous phytohormones and metabolic pro- 2 files in different rhizomatous of Poa pratensis” indicated that cytokinin is crucial for rhizome development and promotes rhizome germination and growth of Poa pratensis.

These authors used modern methods such as UPLC-MS/MS and as well as metabolite profiling.

The subject of the manuscript is consistent with the scope of the Journal. The manuscript is well written and organized in appropriate manner. The conclusions corresponds with the work's content. However, I have found some lacks, for example in description of used methods. I encourage the authors to improve the manuscript according to comments listed below.

Manuscript can be published in scientific Biology  (major revision):

-        Please describe the biological repeats as it was collected?-  chapter  4.3

-        no information on standard solutions of test substances?

-        Please, be sure that all the references cited in the manuscript are also included in the reference list and vice versa with matching spellings and dates.

-        Please increase the quality of the Figures.

Author Response

Dear editor,

We would like to thank you for taking the time to read our manuscript and giving us the opportunity to revise this manuscript. We honestly appreciated the comments, which highlighted the thorough attention to the content, and we tried our best to fulfil them. We have studied all these comments carefully and made corresponding corrections. We respond these comments point by point as below. We hope the new manuscript will meet your magazine’s standard. If any questions still remain, please let us know. We are looking forward to hearing good news from you! Thanks for your attention and patience!

Yours sincerely,

Xiaoming Bai

Comment

The manuscript authors entitled “Differences in endogenous phytohormones and metabolic pro- 2 files in different rhizomatous of Poa pratensis” indicated that cytokinin is crucial for rhizome development and promotes rhizome germination and growth of Poa pratensis. These authors used modern methods such as UPLC-MS/MS and as well as metabolite profiling.

The subject of the manuscript is consistent with the scope of the Journal. The manuscript is well written and organized in appropriate manner. The conclusions corresponds with the work's content. However, I have found some lacks, for example in description of used methods. I encourage the authors to improve the manuscript according to comments listed below.

Manuscript can be published in scientific Biology (major revision)

  1. Please describe the biological repeats as it was collected?- chapter 4.3.

Response: According to your suggestion, we have added this part in the new manuscript. (please line 333-347)

  1. no information on standard solutions of test substances?

Response: Thank you for your reminding. We have added this part of content in our revised manuscript. (please line 333-347)

  1. Please, be sure that all the references cited in the manuscript are also included in the reference list and vice versa with matching spellings and dates.

Response: Thank you very much for your suggestion. We have adjusted the formats and content of references according to the requirements of the journal.

  1. Please increase the quality of the Figures.

Response: Thank you very much for your suggestion. All figures resolution and quality have been improved. Please see them in our revised manuscript.

Round 2

Reviewer 2 Report

Dear Editor in Chief of Biology Journal

Hi and have a good day

I studied the manuscript entitled " Cytokinin and Metabolites Affect Rhizome Growth and Development in Kentucky bluegrass (Poa pratensis) "again and some minor correction is necessary to do by the writers of this article as follows:

 ·         line 159-200: need to edit this paragraph since the word of we used three time in this paragraph. it is not necessary to use we in the MS.

·         Figure 6 shows there is significant difference only for alpha-ketoglutaric acid trait between two genotypes for ABC transporter and Histidine kinase pathways metabolites  via standard error bars was drown on the figures. please check and rewrite again. This problem probably is solvable via data transformation.

·         Please refer to figure 7 in the results section into brackets.

·         line 204-210: significant correlation is important for interpretation. please check and rewrite again this paragraph.

·         line 251-252: please check again the result and explain on bias significant differences.

·         line 377: why FC<0.5 is important? I do not think it is true?

·         line 383-384: please rewrite in this model:

A metabolic pathway was considered enriched when x/n>y/n, and significant when P-value of < 0.05.

·         Finally, The writers of the articles control the other part of the MS such as discussion for corresponding with the result part which will be corrected.

best wishes 

Author Response

Comment

I studied the manuscript entitled " Cytokinin and Metabolites Affect Rhizome Growth and Development in Kentucky bluegrass (Poa pratensis) "again and some minor correction is necessary to do by the writers of this article as follows:

  1. line 159-200: need to edit this paragraph since the word of we used three time in this paragraph. it is not necessary to use we in the MS.

Response: Thank you very much for your suggestion. We have reorganized the section in the revised manuscript. (page 8).

  1. Figure 6 shows there is significant difference only for alpha-ketoglutaric acid trait between two genotypes for ABC transporter and Histidine kinase pathways metabolites via standard error bars was drown on the figures. please check and rewrite again. This problem probably is solvable via data transformation. Please refer to figure 7 in the results section into brackets.

Response: Thank you very much for your suggestion. We have redone the figure according to the requirements again. (page 9).

  1. line 204-210: significant correlation is important for interpretation. please check and rewrite again this paragraph.

Response: Thank you very much for your suggestion. We have reorganized the section in the revised manuscript. (page 9).

  1. line 251-252: please check again the result and explain on bias significant differences.

Response: Thank you very much for your suggestion. We have checked and reorganized the section in the revised manuscript. (page 11).

  1. line 377: why FC<0.5 is important? I do not think it is true?

Response: Thank you very much for your comment. FC means fold change, which is the ratio of the mean value of each metabolite in the comparison group. The default criteria for differential express metabolite screening are FC ≥ 2 or FC ≤ 0.5, VIP > 1, and P value < 0.05. [1-3]

[1] Sreekumar, A. et al. Metabolomic profiles delineate potential role for sarcosine in prostate cancer progression. Nature. 457,910- 914 (2009).

[2] Heischmann, S. et al. Exploratory Metabolomics Profiling in the Kainic Acid Rat Model Reveals Depletion of 25-Hydroxyvitamin D3 during Epileptogenesis. Scientific Reports. 6 (2016).
[3] Haspel, J.A. et al. Circadian rhythm reprogramming during lung inflammation. Nature communications. 5 (2014).
6. line 383-384: please rewrite in this model: A metabolic pathway was considered enriched when x/n>y/n, and significant when P-value of < 0.05.

Response: Thank you very much for your suggestion. We have reorganized the section in the revised manuscript. (page 13).

  1. Finally, The writers of the articles control the other part of the MS such as discussion for corresponding with the result part which will be corrected.

Response: Thank you very much for your suggestion. We have corrected and described the above result section. The objective of this paper is to study the metabolism of phytohormone in rhizomes, mainly explaining the content and role of phytohormone and metabolite compounds in the rhizome.

Reviewer 3 Report

The authors have revised the manuscript thoroughly on the suggested lines and now can be accepted in its current form.

Author Response

Dear editor, 

Thanks very much for taking your time to review this manuscript.  And thank you very much for recognizing our manuscript!

Best Regards,

Reviewer 4 Report

Dear Authors,

The manuscript has been improved.  It its current form, manuscript can be published.

Author Response

(The authors gave the same response as above.)
